# Elexacaftor Mediates the Rescue of F508del CFTR Functional Expression Interacting with MSD2

**DOI:** 10.3390/ijms241612838

**Published:** 2023-08-16

**Authors:** Roberta Bongiorno, Alessandra Ludovico, Oscar Moran, Debora Baroni

**Affiliations:** Istituto di Biofisica, CNR, Via De Marini, 6, 16149 Genova, Italy; bongiorno.roberta@libero.it (R.B.); ale.ludo89@gmail.com (A.L.); oscar.moran@cnr.it (O.M.)

**Keywords:** cystic fibrosis transmembrane conductance regulator (CFTR), cystic fibrosis (CF), F508del CFTR, CFTR correctors, CFTR domains

## Abstract

Cystic fibrosis (CF) is one of the most frequent lethal autosomal recessive diseases affecting the Caucasian population. It is caused by loss of function variants of the cystic fibrosis transmembrane conductance regulator (CFTR), a membrane protein located on the apical side of epithelial cells. The most prevalent CF-causing mutation, the deletion of phenylalanine at position 508 (F508del), is characterized by folding and trafficking defects, resulting in the decreased functional expression of the protein on the plasma membrane. Two classes of small-molecule modulators, termed potentiators and correctors, respectively, have been developed to rescue either the gating or the cellular processing of defective F508del CFTR. Kaftrio, a next-generation triple-combination drug, consisting of the potentiator ivacaftor (VX770) and the two correctors tezacaftor (VX661) and elexacaftor (VX445), has been demonstrated to be a life-changing therapeutic modality for the majority of people with CF worldwide. While the mechanism of action of VX770 and VX661 is almost known, the precise mechanism of action and binding site of VX445 have not been conclusively determined. We investigated the activity of VX445 on mutant F508del to identify the protein domains whose expression is mostly affected by this corrector and to disclose its mechanisms of action. Our biochemical analyses revealed that VX445 specifically improves the expression and the maturation of MSD2, heterologously expressed in HEK 293 cells, and confirmed that its effect on the functional expression of defective F508del CFTR is additive either with type I or type II CFTR correctors. We are confident that our study will help to make a step forward in the comprehension of the etiopathology of the CF disease, as well as to give new information for the development and testing of combinations of even more effective correctors able to target mutation-specific defects of the CFTR protein.

## 1. Introduction

Cystic fibrosis (CF) is an autosomal recessive disease caused by variants in the cystic fibrosis transmembrane conductance regulator (CFTR) protein, a chloride and bicarbonate channel located on the apical side of epithelia, where it plays a critical role in maintaining the electrolyte and fluid balance of the surface liquid layer [1,2]. The lack or dysfunction of CFTR results in thick secretions that cause gastrointestinal, reproductive, and respiratory system defects. Respiratory disorder is the major cause of morbidity and mortality in people with CF. In the airways, a viscous and tenacious mucus constitutes the milieu for chronic inflammation and recurrent infections, leading to epithelial damage, tissue remodeling, and the progressive deterioration of lung function, which ultimately result in respiratory failure [3,4].

As an ABC transporter family member, CFTR contains two transmembrane domains (MSDs), two nucleotide-binding domains (NBDs), and a unique regulatory R region (R domain), which are translated as a single polypeptide from its mRNA transcript [1,5]. CFTR folding is a complex and hierarchical process that takes place in multiple cellular compartments along the secretory pathway. Early during translation, CFTR is inserted in the endoplasmic reticulum (ER), where it is core-glycosylated and folded in a near-native conformation with the assistance of many molecular chaperones. Further maturation continues into the Golgi apparatus, where CFTR becomes fully glycosylated. Finally, mature CFTR is trafficked into secretory vesicles to the plasma membrane (PM) [6,7]. Multiple quality control systems compensate for the limited fidelity of the CFTR biogenesis pathway, recognizing misfolded CFTR and targeting it for ER-associated degradation [8]. Moreover, a system of peripheral protein quality control (PPQR) removes the CFTR protein from the PM if it is recognized as improperly folded [9,10,11,12,13].

Thus far, more than 2000 variants in the CFTR gene have been described (MIM 602421, http://www.genet.sickkids.on.ca/, accessed on 15 June 2023), which give rise to a variety of molecular defects, but, fortunately, only a small number of them produce a clinical phenotype (https://cftr2.org/, accessed on 15 June 2021). CFTR variants have been grouped according to the molecular mechanism that they disturb: protein synthesis mutations (class I), maturation mutations (class II), gating mutations (class III), conductance mutations (class IV), PM abundance mutations (class V), and stability mutations (class VI) [14,15]. Many mutations present pleiotropic defects, which means that they could fit into more than one class. For example, the most prevalent CFTR mutation, the deletion of a phenylalanine at position 508 (F508del) [16,17], is primarily characterized by incomplete folding caused by NBD1 instability (class II), but it presents also gating defects (class III) and reduced cell surface permanence (class VI) [18,19,20,21,22].

The search for CF therapeutics has allowed the identification of two classes of small-molecule modulators able either to improve the permeation of ions through the CFTR pore (namely potentiators), binding directly to mutant CFTR, or to promote the delivery of the processing-defective CFTR protein to the PM (namely correctors), acting as pharmacological chaperones or, eventually, as regulators of proteostasis [23]. Based on their mechanisms of action, correctors have been subdivided into three main classes: type I correctors that target the NBD1/MSD2 interfaces, type II correctors acting on NBD2 or its interfaces, and type III correctors exerting their effect on the folding and stability of NBD1 [24,25]. The current opinion among CF researchers is that the administration of a combination of correctors having a different mechanism of action and a different binding site on the CFTR protein would greatly improve their efficacy in the rescue of defective CFTR.

The correctors lumacaftor (VX809), tezacaftor (VX661), and elexacaftor (VX445) and the potentiator ivacaftor (VX770) are the four CFTR modulators that have successfully progressed through clinical trials and are currently used in the therapy of CF [26,27,28,29]. At present, the most effective therapy for CF is represented by Kaftrio (branded as Trikafta in the USA), consisting of a combination of the potentiator VX770, which increases the open probability of the channel, with the two correctors VX661 and VX445 that synergistically help to drive the processing-defective CFTR from the ER to the PM [30]. The VX770 site of action was found in the MSDs [31], while corrector VX661 is known to exert its effect by enhancing the expression and the stability of MSD1 [32,33,34]; the mechanism of action of corrector VX445 has not been yet fully elucidated. An early study by the Lukacs group [35] showed that VX445 directly binds to NBD1 and stabilizes it, thus classifying this molecule as a type III corrector. Other studies have demonstrated that VX445 is efficient in correcting rare class II mutations residing in different domains of the CFTR, supporting, therefore, the idea that VX445 can promote the multi-domain assembly of the CFTR protein [30,36,37,38,39,40]. It has been also postulated that VX445 exerts dual activity as a defective CFTR corrector and a potentiator [41]. Finally, a recent study by Fiedorczuk and Chen demonstrated that VX445 interacts with helices 10 and 11 of MSD2 and the lasso motif of F508del CFTR [42].

In this work, we applied different biochemical approaches to identify the CFTR domain mainly involved in the rescue of mutant F508del protein expression promoted by VX445. To achieve this aim, we heterologously expressed plasmid constructs encoding the full-length F508del and MSD1, WT and F508del NBD1, the R domain, and the NBD2 and MSD2 isolated domains into highly transfectable human embryonic kidney 293 (HEK-t) cells and analyzed the effect of VX445 on their expression and stability, comparing it with that exerted by other CFTR correctors whose molecular mechanism of action has been previously described [32,33,34,43,44,45]. Our analysis identified MSD2 as the CFTR domain mainly involved in the rescue of F508del CFTR promoted by VX445.

## 2. Results

### 2.1. Effect of Correctors on the Expression of Full-Length WT and F508del CFTR and CFTR Single Domain mRNAs

The relative abundance of constructs encoding full-length WT and F508del CFTR and MSD1, WT or F508del NBD1, the R domain, and the MSD2 and NBD2 single domains was evaluated in HEK-t transfected cells by qRT-PCR, using specific primer pair sets [34,45]. The expression levels of the mRNAs coding for full-length WT and F508del CFTR were not statistically different in untreated or VX809-, VX661-, CORR4A-, and VX445-treated HEK-t cells. Data are presented in Appendix A.

The relative abundance of mRNA extracted from cells transfected with MSD1 was similar in untreated cells and in cells treated with correctors VX809, VX661, CORR4A, and VX445. The relative abundance of WT and F508del NBD1 mRNA in untreated transfected cells was not changed by the treatment with correctors VX809, VX661, CORR4A, and VX445. Similarly, the R domain mRNA yield in untreated cells was not modified by the treatment with correctors. When cells were transfected with NBD2, none of the examined correctors altered the relative abundance of its mRNA. Analogously, treatment with VX809, VX661, CORR4A, and VX445 did not change the expression level of the MSD2 mRNA. Data are presented in Appendix A. In untransfected HEK-t cells, the mRNA encoding full-length CFTR, MSD1, NBD1, R domain, MSD2, and NBD2 was not detected). We concluded that none of the correctors studied in this work modify the mRNA transcription of full-length WT and F508del CFTR or any single CFTR segment.

### 2.2. Effects of Correctors on the Functional Expression of Full-Length WT and F508del CFTR Protein

To test the effects of the VX809, VX661, CORR4A, and VX445 correctors on the expression of full-length WT and F508del CFTR, we treated WT and F508del CFTR transiently transfected HEK-t cells for 24 h with 5 µM VX809, 5 µM VX661, 10 µM CORR4A, 5 μM VX445, or DMSO as a control vehicle. The immunoblot analysis of the retrieved whole-cell extracts is presented in Figure 1. Both CFTR isoforms were detected by the monoclonal antibody MM13-4 raised against the N-terminus of the CFTR protein, as two electrophoretic bands, namely B and C, of approximately 160 and 180 KDa, corresponding to the immature, core-glycosylated, and endoplasmic reticulum (ER) entrapped form and the mature, fully glycosylated, and fully processed CFTR isoform, respectively. As expected, the prevalent band in WT CFTR-transfected HEK-t cell lysates was the C band (first lane of the upper panel of Figure 1A). Lysates of cells expressing F508del CFTR showed primarily the CFTR band B, consistent with the severe folding and trafficking defects caused by the mutation (first lane of the upper panel of Figure 1B). As shown by the graphs in Figure 1A, the treatment of full-length WT CFTR-transfected cells with VX809, VX661, CORR4A, and VX445 did not change either WT CFTR total expression (C + B bands) or its maturation ratio, which was defined as the ratio between the expression of the mature, fully glycosylated (band C) and the total CFTR (B + C bands) forms of the WT CFTR protein. The graphs in Figure 1B show that treatment with VX809, VX661, CORR4A, and VX445 significantly enhanced both F508del CFTR total expression, expressed as the sum of bands B and C, and F508del CFTR maturation, as resulting from the change in the ratio between the expression of the mature, fully glycosylated protein (band C) and the total CFTR protein loading (bands B + C). In particular, VX809 and VX661 demonstrated almost equivalent effects in rescuing the expression of F508del CFTR. Corrector VX445 was the most efficient in rescuing defective CFTR expression, enhancing the expression of defective CFTR by almost three times with respect to the untreated isoform. Appendix A reports the values of WT and F508del CFTR protein total expression, as well as the protein maturation rate, normalized to the intensity of actin, used as a housekeeper protein, and to the expression value of untreated WT or F508del CFTR. For each condition, the statistical significance, as assayed by Dunnett’s multiple comparisons test (all groups against the control group), is also shown.

To verify whether the enhancement in protein expression induced by the treatment with correctors correlated with an increase in defective CFTR channel activity, we measured the transport of iodide mediated by the F508del CFTR protein permanently transfected into Fisher rat thyroid (FRT) and human bronchial epithelial F508del CFBE41O^−^ (CFBE41O^−^) cells expressing the iodide-sensitive protein YFP. The iodide influx mediated by CFTR channels was measured as the initial quenching rate (QR) of the YFP fluorescence. Figure 2A–D show that the treatment with correctors elicited a significant augmentation in the iodide influx either in FRT or CFBE41O^−^ cells. In particular, VX445 elicited in FRT and CFBE41O^−^ cells an increase in the QR that was 2.61 and 2.52 times higher than that of the control condition, respectively (Figure 2).

### 2.3. Effect of Correctors on the Expression of CFTR Single Domains

To verify whether the treatment with the compounds under study could determine an increase in the expression of any single domain of the CFTR protein, we transiently transfected HEK-t cells with MSD1, WT and F508del NBD1, R domain, MSD2, and NBD2 and successively treated them with 5 μM VX809, 5 μM VX661, 10 μM CORR4A, and 5 μM VX445 for 24 h. Immunoblots of whole-cell lysates from CFTR single domain transfected HEK-t cells are shown in the upper panels of Figure 3A–F. MSD1 was revealed by the MM13-4 antibody as an electrophoretic band of ~45 kDa, while both WT and F508del NBD1 polypeptides were detected as bands of ~32 kDa by the L12B4 antibody. The R domain had an apparent molecular weight of ~22 kDa. The primary antibodies raised against the MSD2 and NBD2 domains revealed these polypeptides as bands of ~47 and ~30 kDa, respectively. Controls in untransfected HEK-t cells showed that no CFTR single domain was detected in the blots by the primary antibodies that were used. Appendix A summarizes the characteristics of the primary antibodies used in this work.

Treatment with VX809 and VX661 increased the expression of MSD1. Indeed, in cells treated with these correctors, the MSD1 expression was 2.69 and 2.61 times higher than in control DMSO-treated cells, respectively. On the contrary, CORR4A and VX445 did not modify the expression of MSD1 (Figure 3A). No significant increase in NBD1 protein expression was observed in WT- and F508del NBD1-transfected cells upon treatment with VX809, VX661, CORR4A, or VX445, respectively (Figure 3B,C). Similarly, the correctors under study did not elicit any significant effect on the expression level of the R domain (Figure 3D). The expression of MSD2 was not changed upon treatment with correctors VX809, VX661, or CORR4A (Figure 3E). On the contrary, VX445 increased the expression of this domain to 2.88 times that observed in control untreated HEK-t cells. The expression of NBD2 was significantly increased in transfected HEK-t cells treated with CORR4A (Figure 3E). Indeed, upon treatment with this corrector, the NBD2 protein expression in HEK-t whole-cell lysates was 2.34 times higher than in control DMSO-treated HEK-t cells. Treatment with VX809, VX661, or VX445 did not modify the expression level of NBD2. The expression values of heterologously transfected CFTR single domains together with the outcomes of the statistical analyses are presented in Appendix A.

### 2.4. Effect of Tested Correctors on the Stability of NBD2

As VX445 seemed to mostly affect the expression of MSD2, we assessed whether this compound could exert an effect on the stability of this polypeptide. To allow for a comparison, the impact of the treatment with VX809, VX661, CORR4A, and VX445 on the MSD2 polypeptide’s half-life was also assayed (Figure 4). Cycloheximide chase experiments showed that the expression of MSD2 in control DMSO untreated samples decayed to 41% after 4 h and to 30% after 6 h from the beginning of the treatment with cycloheximide (Figure 4A). Analogously, the expression level of MSD2 treated with VX809 and VX661 was 42% and 47% and 31% and 33% of the initial one after 4 and 6 h from the blockade of protein synthesis with cycloheximide, respectively (Figure 4B,C). The treatment with CORR4A did not modify MSD2’s stability, as the expression level of MDS2 was 32% and 25% of the initial one after 4 and 6 h from the beginning of the treatment with cycloheximide (Figure 4D). Contrarily, VX445 determined a significant increase in MSD2’s stability. In fact, in VX445-treated samples, the expression level of MSD2 was 59% and 44% of the initial one after 4 and 6 h from the beginning of the treatment with cycloheximide, respectively (Figure 4E). Appendix A reports, for each time interval depicted in Figure 4, the values of MSD2 expression, normalized to the intensity of actin, used as a housekeeper protein, and to the expression of MSD2 at the beginning of the treatment with cycloheximide. For each condition, the statistical significance, as assayed by Dunnett’s multiple comparisons test (all groups against the control group), is also shown.

### 2.5. Effect of Combinations of Correctors on the Channel Activity of Full-Length F508del CFTR Protein

We then investigated the possible effect of combinations of correctors on the channel activity of the F508del protein by means of the YFP-based assay. Such experiments were performed on FRT and CFBE41O^−^ cells, stably expressing the F508del and the YFP proteins. As already asserted, cells were incubated with a cocktail consisting of forskolin and VX770 to stimulate the maximal F508del channel activity. We observed that VX445 generated significant additive/synergistic effects when combined with the VX809, VX661, and CORR4A correctors (Figure 5A–D). In particular, the VX445 + VX809 double corrector combination generated a higher increase in the QR, which was 3.19 and 3.11 times that elicited by VX445 alone, in FRT and CFBE41O^−^ cells, respectively. Moreover, the double corrector combinations VX445 + CORR4A, CORR4A + VX809, and CORR4A + VX661 caused a significant increase in F508del-mediated I^−^ transport, while the effect on F508del channel activity exerted by the combination VX809 + VX661 was not different from that elicited by VX445 alone. We then evaluated the effect of triple combinations of correctors on F508del function. In both cell lines, two out of three triple combinations, VX445 + VX809 + CORR4A and VX445 + VX661 + CORR4A, determined a significant enhancement in the QR with respect to VX445 alone. On the contrary, the F508del-mediated I^−^ transport elicited by the triple combination VX445 + VX809 + VX661 was not different from that prompted by VX445 alone. None of the triple combinations were more effective than each of the double combinations used in this study in enhancing the QR (Figure 5A–D).

## 3. Discussion

In recent years, a great deal of effort has been devoted to the development of therapeutics able to correct and improve the expression, processing, and function of defective CFTR protein (CF drug development pipeline, htps://www.cff.org/trials/pipeline (accessed on 10 August 2023)). The triple combination Kaftrio (Trikafta), including the potentiator ivacaftor (VX770) and the two correctors tezacaftor (VX661) and elexacaftor (VX445), constitutes today the most relevant outcome achieved by researchers in the field of CF [35,36,37,38,39,40,41,42]. Demonstrating clinical benefits greater than those of its single components, Kaftrio has received FDA and EMA approval for administration to patients with CF that are F508del homozygotes or F508del heterozygotes with a residual function mutation, who account for the 90% of those with CF [46].

Besides the effect on CFTR gating, due to the potentiator VX770, the therapeutic effect of Kaftrio is expected to arise from the synergistic action of the two correctors, VX661 and VX445, which are supposed to have complementary mechanisms of action on the maturation and traffic of CFTR. Cryo-EM analysis pointed out that the VX770 site of action resides in the transmembrane domain [31]. VX661 belongs to type I correctors, which are believed to support and stabilize the formation of the NBD1/MSD1 and NBD1/MSD2 interfaces, by directly interacting with MSD1 [32,33,34]. VX445 has been discovered only recently and its mechanism of action has not been yet completely elucidated. Early functional studies showed that VX445 has additivity/synergy with type I correctors [35]. In particular, the functional YFP assay showed that the combination of VX445 with either VX809 or VX661 was particularly effective in primary bronchial epithelial cells derived from patients with CF [36,37,38,39]. Other studies proposed that VX445 acts as a potentiator [47] or plays a dual role as a potentiator and a corrector [41,48]. Finally, in a recent study, Fiedorczuk and Chen resolved the molecular structure of F508 CFTR bound to the three components of Kaftrio and identified the pocket formed by the transmembrane helices 10 and 11 and the lasso motif as the putative site of interaction of VX445 with F508 CFTR [42]. In this work, we aimed to elucidate the mechanism by which VX445 acts to rescue the expression of the F508del CFTR protein. To achieve this aim, we analyzed the effect of VX445 on the expression of whole-length F508del CFTR and on the different domains of the CFTR protein, singly transfected in HEK-t cells, a heterologous expression system widely used for its higher transfectability. As the CFTR domains whose expression is mostly affected by type I and type II correctors are almost known [32,33,34,43,44,45] we used VX809, VX661, and CORR4A CFTR correctors as reference compounds to compare the obtained results.

Before proceeding with the biochemical evaluation of protein expression, we preliminarily ascertained whether VX445 as well as VX809, VX661, and CORR4A elicited any effect on the expression of the mRNA of the plasmid constructs that we used in our study. The expression levels of the transcripts of WT and F508del CFTR and of MSD1, WT and F508del NBD1, and the NBD2 and MSD2 domains transfected in HEK-t cells were evaluated by real-time PCR (see Appendix A). The RNA coding for all the constructs that we used was approximately constant in all preparations, independently of the treatment (or not) with correctors; therefore, we concluded that, similarly to VX809, VX661, and CORR4A [34,45], VX445 exerted its effect on the expression of the CFTR constructs at the post-transcriptional level.

Then, we analyzed the expression levels of the WT and F508del CFTR total (B + C bands) and mature (C/(C + B) band ratio) forms in SDS-PAGE whole-cell lysates obtained from control untreated and corrector-treated samples. Our analysis showed that in WT CFTR lysates, neither total nor mature WT CFTR protein increased after treatment with VX809, VX661, CORR4A, or VX445 (Figure 1A). On the contrary, all correctors under study were demonstrated either to enhance the total expression or promote the maturation of F508del CFTR (Figure 1B). Specifically, the next-generation corrector, VX445, exerted a stronger effect among the correctors that we tested (Figure 1B). It is noted that, thus far, the achieved results do not allow us to distinguish whether the correctors directly act by increasing F508del CFTR biogenesis/production or, alternatively, modulate any component of the cellular machinery responsible for defective protein recognition and degradation, such as the ERAD.

In the next step of our analysis, we tested the capability of VX4455 to influence the F508del CFTR-mediated iodide transport and compared it with that of the other correctors used in this study. Such experiments were conducted in parallel in FRT and CFBE41O^−^ cells stably expressing the YFP and the F508del CFTR. Analogously to what emerged from our analysis of whole-length F508del protein expression, the iodide transport of FRT cells was greatly reduced (Figure 2A,C), even if F5089del channel activity was maximally stimulated either by VX770 or forskolin. Similar results were obtained in CFBE41O^−^ cells (Figure 2B,D). In both YFP and F508del transfected cell lines, the channel transport activity of the corrector-treated F508del CFTR followed the order VX445 > VX809 > VX661 > CORR4A. The achieved results are in agreement with Ussing chamber recordings of transepithelial currents elicited in CFBE41O^−^ F508del cells [36] and indicate that the amelioration of the processing defects exerted by correctors positively correlates with the increase in F508del channel activity.

The pivotal point of our study focused on the identification of CFTR domains that are mainly affected by the action of VX445. As already stated, to achieve this goal, we generated expression constructs containing CFTR single domains: MSD1 (M1, residues 1–388), NBD1 (N1, residues 348–633) in both WT and F508del isoforms, R (residues 645–834), MSD2 (M2, residues 837–1218), and NBD2 (N2, residues 1210–1480) [34,45]. We used two type I correctors, VX809 and VX661, and a type II corrector, CORR4A, as reference compounds to allow for comparison. As highlighted by Figure 3, results regarding correctors VX809, VX661, and CORR4A confirmed the outcomes already obtained by ours and other groups [32,33,34,43,44,45]. In particular, corrector VX809 and its derivative VX661 increased the expression of MSD1, while they demonstrated an almost negligible effect on the expression levels of the other CFTR domains. Analogously, NBD2 in the CFTR domain was mainly involved in the interaction between mutant F508del CFTR and CORR4A. Regarding VX445, our results indicated that MDS2 was the region most affected by the action of this corrector (Figure 3). Indeed, VX445 significantly increased the expression of this domain, while it was unable to enhance the expression of the other singly expressed CFTR domains. The retrieved results confirm the observations obtained by means of Cryo-EM by Fiedorczuk and Chen [42]. Indeed, these authors identified a hydrophobic pocket constituted by transmembrane helices 10 and 11 of MSD2 and the lasso motif as the subdomains most extensively involved in the interaction between VX445 and F508del CFTR.

Another aspect of the interaction between VX445 and F508del that we sought to address in this work was whether VX445 had an effect on the stability of MSD2. Corroborated by previous results obtained with other CFTR modulators [34,45], we hypothesized that if VX445 acts by stabilizing the region whose expression it increases, then it could be expected that this region would have a slower turnover rate after transiently transfected cells are treated with cycloheximide to inhibit protein synthesis. Therefore, HEK-t cells were transfected with plasmids encoding MSD2 and incubated in the presence of VX445, VX809, VX661, CORR4A, or DMSO. The next day, protein synthesis was inhibited by the addition of cycloheximide. Whole-cell extracts were collected at various time points and subjected to SDS-PAGE and Western blot analysis. In the absence of correctors, MSD2 expression decayed to 41% and 25% after 4 and 8 h, respectively. As expected, VX445 significantly prolonged the half-life of MSD2, while the other correctors failed to exert this type of action on MSD2 (see Appendix A and Figure 4). In fact, in VX445-treated samples, the expression level of MSD2 was 59% and 44% of the initial one after 4 and 6 h from the beginning of the treatment with cycloheximide, respectively (Figure 4E). Only one issue remains unresolved: VX445 exerted no effect on MSD2 abundance in samples incubated for 8 h with cycloheximide. At present, we do not have a mechanistic explanation for this fact. We can only state that we observed high mortality in all cell preparations (not only in those treated with VX445), and that surviving cells were also suffering. It is possible that the blockade of protein synthesis achieved by long-term incubation with cycloheximide may have compromised mechanisms that are fundamental not only for the survival of cells but also for their proper metabolism. However, taken as a whole, the achieved results further strengthen the notion that MSD2 is the region whose expression is most affected by VX445.

As Kaftrio, the revolutionary therapeutic that has radically improved the lives of many persons affected by CF, is a combination of CFTR modulators, specifically VX445 with corrector VX661 and potentiator VX770, we tested VX445’s capability to increase F508del CFTR channel activity when administrated in combination with the other correctors used in this study. VX661 and VX809 belong to type I correctors, which are believed to support the formation of the NBD1/MSD1 and NBD1/MSD2 interfaces, by directly binding to MSD1 [32,33,34,45]. From our findings, it emerges that VX445 acts to enhance MSD2 expression and stability (Figure 3 and Figure 4). Consequently, VX445 and type I correctors should interact with different regions of mutant CFTR and their combined activity should help to rescue more than one of the different defects of the F508del protein. Interestingly, when we investigated the correction efficacy of VX445 in combination with VX809 or VX661, we observed that in FRT and CFBE41O^−^ cells, the efficacy of the VX445 + VX809 and VX445 + VX661 combinations was higher than that of each corrector administrated singly (Figure 5A–D). This notable result, which explains the clinical benefits of Kaftrio, has been also observed in heterologous expression systems and primary airway epithelial cells by means of biochemical and functional assays [36,37,38,39,40]. Our results also pointed out that VX445 displayed additivity/synergy with CORR4A, a CFTR corrector categorized as a type II corrector [43,44]. The enhancement in the F508del channel function obtained with this double corrector combination was comparable to that obtained by VX445 in combination with type I correctors (Figure 5A–D). On the other hand, also the combination of a type I, VX809 or VX661, and CORR4A, a type II corrector, produced a significant increase in F508del channel activity. Similar behavior was not observed when two correctors of the same type, specifically VX809 and VX661, were used in combination (Figure 5A–D). Given that the primary strategy by which to definitively treat the CF disease is to use personalized combinations of CFTR modulators able to specifically rescue the molecular and cellular defects of the specific CF-causing mutation, as the last task of our work, we analyzed the effect of VX445 administrated in a triple combination with the other correctors used in this study on the transportation capability of VX770 and forskolin-stimulated FRT and CFBE41O^−^ cells. Surprisingly, none of the triple combinations assayed evoked an effect stronger than that elicited by the double corrector combinations in enhancing F508del-mediated iodide transport (Figure 5A–D). As the drugs used in this work were all used at saturating concentrations [35], this unexpected outcome could be due to the allosteric inhibition exerted by one (or more) modulator(s) on the correct engagement of the other CFTR modulators to their target domains. Given the wide variety of different CFTR drug combinations, and the possibility to improve the chemical properties and structures of existing lead compounds, we plan to study in more detail the potential of triple combinations of CFTR modulators and to identify the domains and subdomains of CFTR most affected by their action, applying our biomolecular approaches to even smaller CFTR subdomains. We are confident that our efforts will allow us to enhance the understanding of the molecular and cellular mechanisms that are affected by the multiple defects that cause CF diseases, as well as to provide new information for the development of personalized combinations of CFTR modulators able to elicit a real clinical benefit for people with CF bearing class II and III mutations.

In summary, there are two main considerations that emerge from our work. First, although we have not provided any direct evidence that correctors directly bind to the F508del CFTR, our findings indicate that correctors specifically influence the expression and the stability of different regions of F508del CFTR—specifically, correctors VX809 and VX661 in MSD1, corrector CORR4A in NBD2, and the next-generation corrector VX445 in MSD2. Second, although all correctors tested in this study demonstrated an ability to increase the functional expression of the mutant CFTR isoform when administrated alone, the use of drug combinations represents the best choice to correct the multiple defects that affect F508del CFTR.

## 4. Materials and Methods

### 4.1. Chemicals

The 3-[6-[[[1-(2,2-difluoro-1,3-benzodioxol-5-yl)cyclopropyl]carbonyl]amino]-3-methyl-2-pyridinyl]-benzoic acid (Lumacaftor, VX809), 1-(2,2-difluoro-1,3-benzodioxol-5-yl)-N-[1-[(2R)-2,3-dihydroxypropyl]-6-fluoro-2-(2-hydroxy-1,1-dimethylethyl)-1H-indol-5-yl]-cyclopropanecarboxamide (Tezacaftor, VX661), N-[2-(5-chloro-2-methoxyphenylamino)-2 0-yl]benzamide (CORR4A), and (S)-N-((1,3-dimethyl-1H-pyrazol-4-yl)sulfonyl)-6-(3-(3,3,3-trifluoro-2,2-dimethylpropoxy)-1H-pyrazol-1-yl)-2-(2,2,4-trimethylpyrrolidin-1-yl)nicotinamide (Elexacaftor, VX445) correctors were purchased from Selleck Chemicals (Munich, Germany). If not explicitly indicated in the text, all other chemicals and culture media components were provided by Merck (Milan, Italy).

### 4.2. Cell Culture

Human highly transfectable embryonic kidney 293 (HEK293-t) cells were purchased from the Interlab Cell Line Collection (Genoa, Italy). Cells were grown in Dulbecco’s modified Eagle’s medium (DMEM) supplemented with 2 mM L-glutamine, 1% PenStrep (100 U/mL), and 20% FBS, at 37 °C and 5% CO_2_. To prevent the loss of differentiation potential, cells were not allowed to become confluent. Fisher rat thyroid (FRT) and human bronchial epithelial F508del CFBE41O^−^ (CFBE41O^−^) cells stably co-transfected with a halide-sensitive yellow fluorescent protein (YFP-H148Q/I152L [49,50]) and F508del-CFTR were cultured in Coon’s modified and MEM media, respectively. In both cases, media were supplemented with 10% FBS, 2 mM L-glutamine, 1% PenStrep (100 U/mL), and 1 mg/mL geneticin (G418) and 0.6 mg/mL zeocin as selection agents at 37 °C and 5% CO_2_.

### 4.3. Generation and Expression of CFTR Constructs

Plasmids coding for whole WT and F508del CFTR molecules (residues 1–1480), as well as those encoding MSD1 (residues 1–388M), NBD1 (residues 348–633), MSD2 (residues 837–1218), and NBD2 (residues 1210–1480), were subcloned between Hind III and XhoI, and the construct coding for the R domain (residues 645–834) between the Hind III and EcoRI restriction sites of the expression vector pCDNA3 (Invitrogen, Paisley, UK) [34,45]. The cDNAs encompassing phenylalanine at position 508 of the CFTR molecule were further modified by site-directed mutagenesis to introduce the F508del deletion, using a QuickChange kit (Stratagene, Santa Clara, CA, USA). The mutation was verified by sequencing (Biofab Research, Rome, Italy).

For transfection, HEK-t cells were plated onto poly-L-lysine-coated culture dishes and grown to 65% confluence in a complete medium. Cells were transiently transfected using Lipofectamine 2000 (Invitrogen, Paisley, UK) with 4 μg of cDNA. The transfection medium (DMEM supplemented with 2 mM L-glutamine and without FBS) was replaced after 6 h with a fresh complete medium containing 5 µM VX809, 5 µM VX661, 10 µM CORR4A, 5 μM VX445, or vehicle DMSO (control). Cells were harvested after 24 h.

### 4.4. RNA Isolation, Reverse Transcription, and Quantitative Real-Time Polymerase Chain Reaction

Total RNA was isolated using the RNeasy Mini Kit (Qiagen, Hilden, Germany) and first-strand cDNA was synthesized from 2 µg of RNA using the RevertAid First Strand cDNA Synthesis Kit and random hexamers according to the manufacturer’s instructions (Fermentas, Burlington, ON, Canada). First-strand cDNA from transfected HEK293-t cells was employed as the template in a quantitative real-time polymerase chain reaction (qRT-PCR) in a CFX Connect Real-Time PCR Detection System instrument (Bio-Rad Laboratories, Hercules, CA, USA). The sequences of the oligonucleotide primer pairs specific to full-length CFTR, MSD1, NBD1, R-domain, MSD2, NBD2, and glyceraldehyde-3-phosphate-dehydrogenase (GAPDH), used as a housekeeper gene, and the amplification conditions, are listed elsewhere [34]. Changes in cDNA amounts were evaluated using the comparative cycle threshold (Ct) method. Each sample was run at least in quadruplicate.

### 4.5. Western Blot

Cells were lysed in a RIPA lysis buffer (50 mM Tris–HCl, pH 8.0, 150 mM NaCl, 1% Triton X-100, 1% sodium deoxycholate, 0.1% SDS) containing a complete protease inhibitor cocktail (Merk, Milan, Italy. The protein concentration was determined by Bradford’s method using bovine serum albumin as the standard. Equal amounts of protein (30 μg) were subjected to SDS-PAGE and transferred to a PVDF membrane (Millipore, Billerica, MA, USA). Blots were incubated with primary antibodies raised against different domains of the CFTR protein, whose characteristics are summarized in Appendix A. Goat anti-mouse or anti-rabbit horseradish peroxidase-conjugated antibodies (dilution 1:2000; Santa Cruz Biotechnologies, Dallas, TX, USA) were used as secondary antibodies. Immuno-detection was performed using Amersham ECL PLUS detection reagents (GE Healthcare Europe GmbH, Milan, Italy), and the images were captured using Amersham Hyperfilm ECL. To confirm the homogeneity of the loaded proteins, immunoblots were stripped by incubating them with stripping buffer (62.5 mM Tris–HCl, pH 6.8, 10% SDS, and 1% β-mercaptoethanol) for 30 min at 55 °C and reprobing them with an anti-actin polyclonal antibody (1:2000, Sigma). Untransfected cell lysates, used as negative controls, were assayed with anti-CFTR and anti-actin antibodies. For quantification, densitometry of the Western blot images was performed with the ImageJ software version 1.53t (U.S. National Institutes of Health, Bethesda, MD, USA). For each lane, the bands, analyzed as regions of interest, were quantified and normalized to the intensity of the band corresponding to actin detected in the stripped PVDF membranes. The Western blot of each analyzed condition was repeated at least in four independent experiments.

### 4.6. Cycloheximide Chase Assay

To evaluate the stability of the MSD2 polypeptide, the HEK-t cells were transfected with the plasmid containing the cDNA encoding this construct and incubated for 24 h in the presence of DMSO (control), 5 µM VX809, 5 µM VX661, 10 µM CORR4A, and 5 μM VX445. Protein synthesis was then inhibited by the addition of 0.5 mg/mL cycloheximide. Cells were harvested at six different time points (after 0, 1, 2, 4, 6, and 8 h), and samples of whole-cell SDS extracts were subjected to immunoblot analysis.

### 4.7. YFP-Based Assay

For the YFP-based assay, FRT or CFBE41O^−^ cells stably co-transfected with a halide-sensitive yellow fluorescent protein (YFP-H148Q/I152L; [49,50]) and F508del CFTR were cultured in standard conditions (37 °C, 5% CO_2_) on black-wall, clear-bottom, 96-well micro-plates at a density of 30,000 cells per well. The day after seeding, cells were incubated for 18 h with 5 µM VX809, 5 µM VX661, 10 µM CORR4A, and 5 μM VX445 or DMSO as a control. Moreover, double or triple combinations of the aforementioned correctors were assayed. The assay is based on the fact that the YFP protein fluorescence is quenched to a greater extent by I^−^ than by Cl^−^ [49,50]. The influx of iodide mediated by the transport activity of the CFTR channel was measured using a fluorescence plate reader (Tristar2 S, Berthold Technologies, Bad Wildbad, Germany) equipped with 485 nm excitation and 535 nm emission filters. If not otherwise stated, 40 min before the assay, the cells were washed twice with a solution containing (in mM) NaCl 136, KNO_3_ 4.5, Ca(NO_3_)_2_ 1.2, MgSO_4_ 0.2, Glucose 5, HEPES 20 (pH 7.4). The cells were then incubated in 60 μL of this solution at 37 °C for 25 min with 20 µM of forskolin and 1 µM VX770 to maximally stimulate the CFTR activity.

Once the assay started, the fluorescence was recorded every 0.2 s for as long as 25 s for each well. Two seconds after the beginning of the recording of the fluorescence, 100 μL of an extracellular solution containing 136 mM NaI instead of NaCl was injected, so that the final concentration of NaI in the wells was 85 mM. The iodide influx was detected as fluorescence quenching as the I^−^ anion bound to the intracellular YFP. The initial rate of fluorescence decay (QR) was derived by fitting the signal with an exponential function, after background subtraction and normalization for the average fluorescence before NaI addition.

### 4.8. Statistics

Data were analyzed using the Igor Pro software (version 9.0.2.4, Wavemetrics, Lake Oswego, OR, USA). Results are expressed as mean ± SEM (standard error of the mean). Dunnett’s post hoc multiple comparisons test was run after a significant one-way analysis of variance (ANOVA) to compare data sets. In all cases, significance was accepted for a probability of *p* < 0.05.

## Figures and Tables

**Figure 1 ijms-24-12838-f001:**
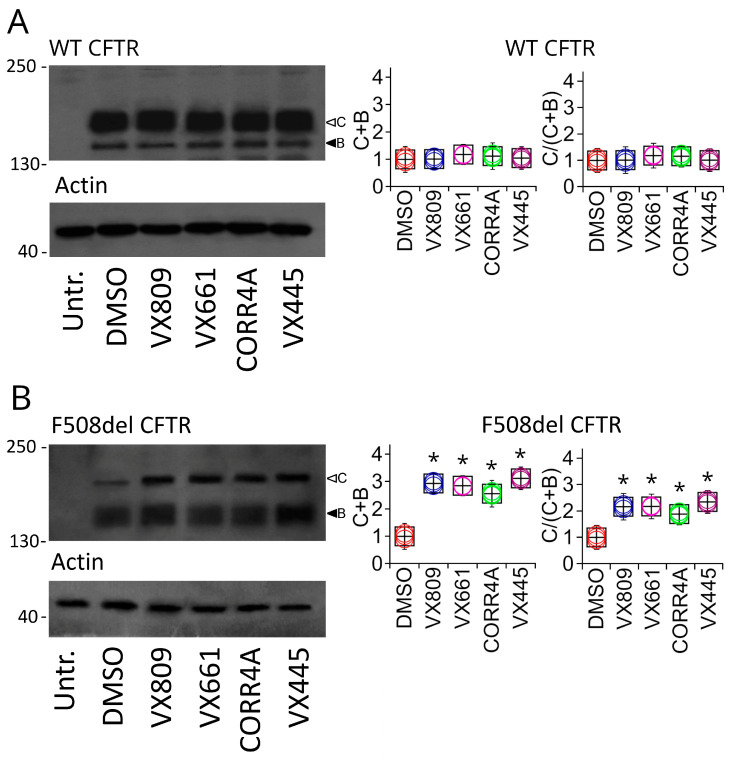
Detection of full-length WT and F508del CFTR proteins by Western blot. Detection of WT (**A**) and F508del CFTR (**B**) in lysates of untransfected (untr.) and transiently transfected HEK-t cells treated with DMSO (control), VX809, VX661, CORR4A, and VX445 is shown in the upper panels on the left. Expression of the housekeeper protein actin in the same samples is shown in the lower panels. The molecular weight of the proteins of the molecular weight marker that was run in the SDS-PAGE is indicated on the left of each blot. Black and white arrowheads indicate the positions of bands B and C, respectively. Bar graphs showing the quantification of the expression of total (calculated as the sum of bands B and C) and mature, fully glycosylated (expressed as C/(C + B) ratio) WT or F508del CFTR isoforms are shown in the middle and on the right of the figure. The expression level of each band was normalized to the level of actin detected in the same samples and expressed relative to the expression level of the control sample. For each condition, the black boxes represent the mean ± standard error of the mean (sem), while colored circles are the single measurements. At least 4 independent experiments were performed. Statistical comparison of the data was performed by Dunnett’s multiple comparisons test (all groups against the control group). Asterisks indicate statistical significance versus control, DMSO-treated samples: * *p* < 0.05.

**Figure 2 ijms-24-12838-f002:**
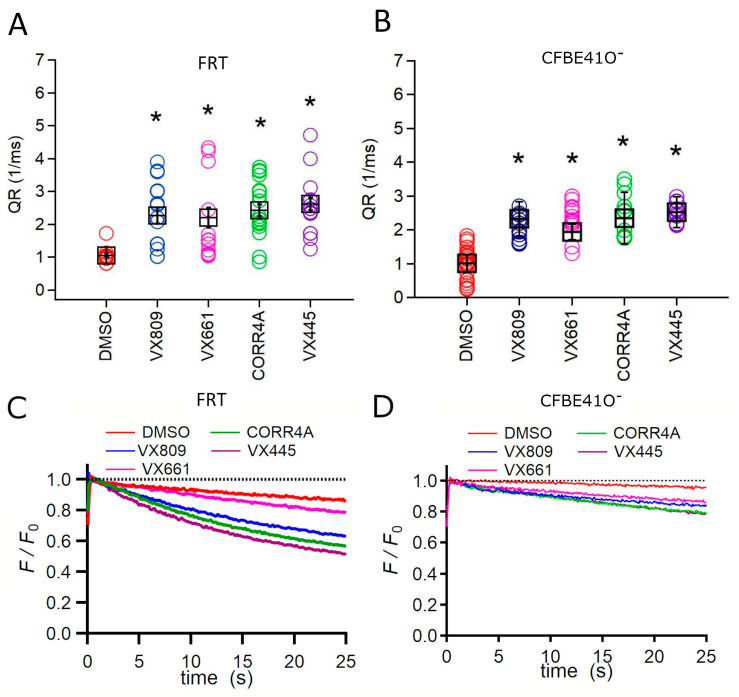
Evaluation of F508del CFTR channel function. Channel activity expressed as the initial fluorescence quenching rate (QR) of the yellow fluorescent protein (YFP) elicited by the iodide influx in FRT (**A**) and CFBE41O^−^ (**B**) cells permanently co-transfected with the YFP and the F508del CFTR proteins. Time course of the fluorescence decay in FRT (**C**) and CFBE41O^−^ (**D**) cells incubated with the correctors under study. The fluorescence was normalized to the initial value obtained after the addition of iodide. Cells were treated with DMSO, 5 μM VX809, 5 μM VX661, 10 μM CORR4A, and 5 μM VX445. For each condition, the black boxes represent the mean ± standard error of the mean (sem), while colored circles are the single measurements. For each condition, each measurement (n) was repeated at least 10 times. Asterisks indicate a significant difference (* *p* < 0.05) compared to the control, DMSO-treated samples.

**Figure 3 ijms-24-12838-f003:**
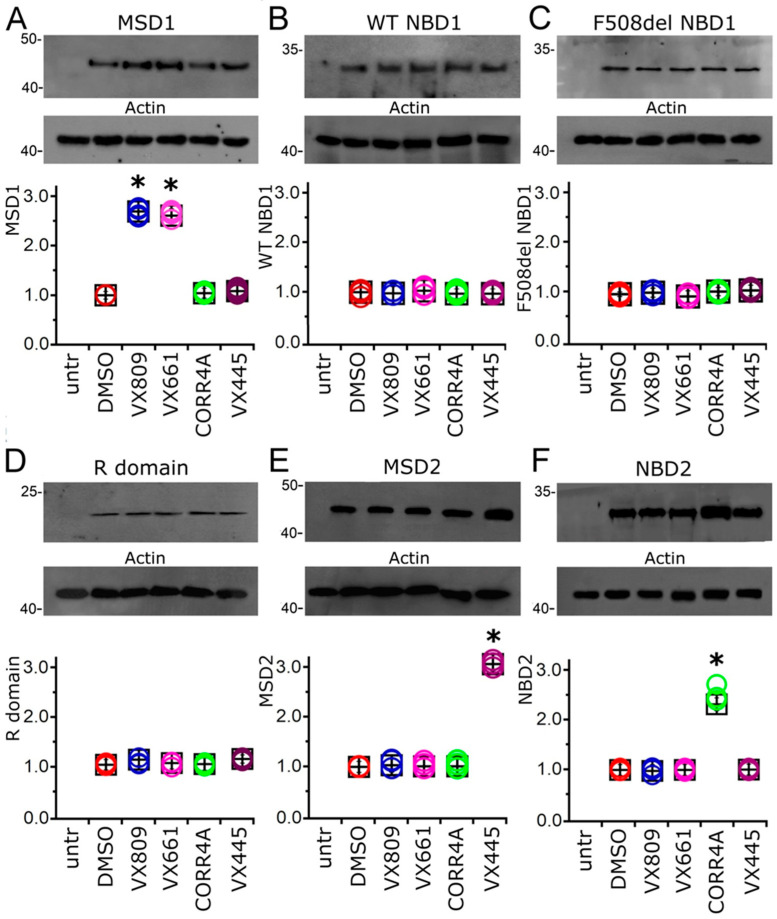
Effect of correctors VX809, VX661, CORR4A, and VX445 on the expression of CFTR single domains. Western blots of MSD1 (**A**), WT (**B**), del508F NBD1 (**C**), R domain (**D**), MSD2 (**E**), and NBD2 (**F**). CFTR domains in untransfected (untr) and in transiently transfected in HEK-t cells treated with DMSO (control) or with 5 μM VX809, 5 μM VX661, 10 μM CORR4A, and 5 μM VX445, respectively. In the lower blots, we show the expression of actin, used as a housekeeping protein. The molecular weight of the protein of the molecular weight marker that was run in the SDS-PAGE is indicated on the left of each blot. The bar graphs at the bottom of each panel indicate the normalized expression level of each single domain. Data are expressed as mean ± standard error of the mean (sem) of at least 4 independent experiments. Dunnett’s test was used for data comparison. Asterisks indicate statistical significance versus control: * *p* < 0.05.

**Figure 4 ijms-24-12838-f004:**
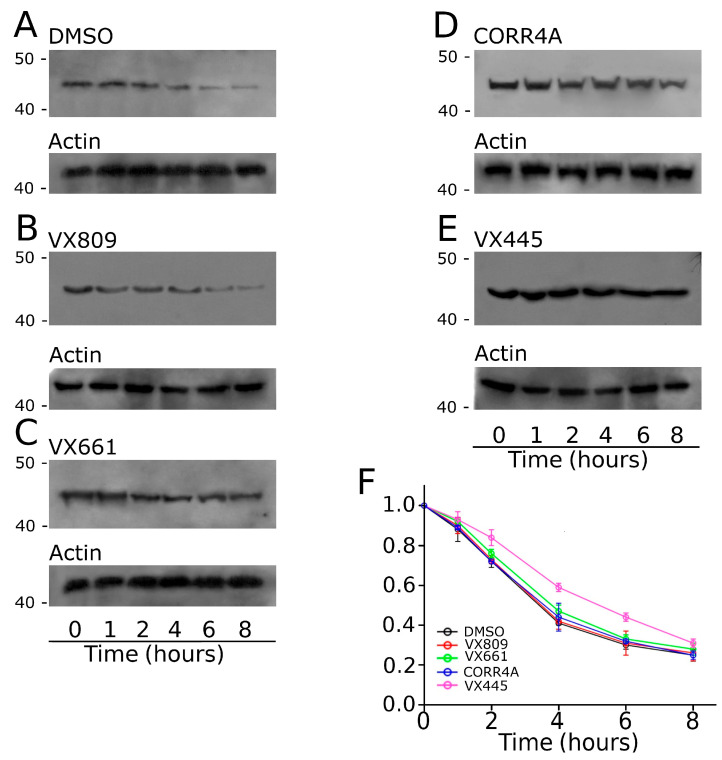
Evaluation of MSD2 stability by means of the cycloheximide chase approach. Expression of the MSD2 polypeptide in HEK-t cell lysates treated with DMSO (**A**), 5 µM VX809 (**B**), 5 µM VX661 (**C**), 10 µM CORR4A (**D**), and 5 µM VX445 (**E**) and subjected to protein synthesis inhibition by means of incubation with 0.5 mg/mL cycloheximide. The lanes of each blot represent 6 different time points as indicated at the bottom of the figure. For each condition, the expression of the protein actin is shown in the lower panels of (**A**–**E**), respectively. The molecular weight of the protein of the molecular weight marker that was run in the SDS-PAGE is indicated on the left of each blot. (**F**) Expression of the MSD2 protein at each time point. Data are expressed as means ± SEM of at least 4 independent experiments. In the legend of the figure, we indicate the symbols used to represent each compound used in the experiment. For all conditions under analysis, the amount of the MSD2 protein was normalized to actin and expressed relative to time 0.

**Figure 5 ijms-24-12838-f005:**
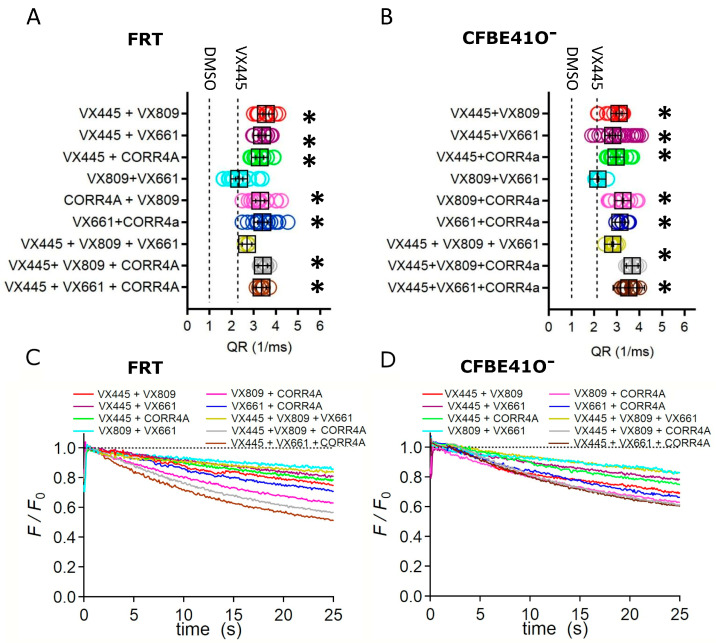
Evaluation of corrector combinations’ effects on F508del channel function. (**A**) Quenching rate (QR) of the yellow fluorescent protein (YFP) elicited by the iodide influx in FRT cells permanently co-transfected with YFP and the F508del CFTR proteins. (**B**) Time course of the fluorescence decay in FRT (**C**) and CFBE41O^−^ (**D**) cells incubated with the correctors under study. The fluorescence was normalized to the initial value obtained after the addition of iodide. Cells were treated with double or triple combinations of the correctors tested in this work. In both figures, the black boxes represent the mean ± standard error of the mean (SEM), while colored circles are the single measurements. The dashed lines represent the QR elicited in the cell preparations incubated with DMSO or VX445, respectively. For each condition, each measurement (n) was repeated at least 10 times. Asterisks indicate a significant difference (* *p* < 0.05) with respect to VX445-treated samples.

## Data Availability

The data presented in this study are available on request from the corresponding author.

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
