# Peer review of "Elexacaftor Mediates the Rescue of F508del CFTR Functional Expression Interacting with MSD2"

_ijms, 2023, doi:10.3390/ijms241612838_

Round 1
Reviewer 1 Report
The research manuscript “Elexacaftor Mediates the Rescue of F508del CFTR Functional Expression Interacting with the MSD2” presents interesting experiments done to elicit the molecular target of VX445, one part of the triple combination Trikafta. The experiments were performed in recombinantly expressed CFTR and the isolated domains and the results are carefully presented. The statistical analysis is correctly performed, the references are accurate and the manuscript is of general interest. Particularly interesting is the differential effect of the different correctors and potentiators on different domains of CFTR. Specifically, the correctors VX809 and VX661 affect MSD1, corrector CORR4A affects NBD2 and VX445 affects MSD2. Mainly, the concerns with the manuscript are language-related.
Concerns which need correction:
1: There are several sentences with unclear meaning, for example:
Line 53: Among these cellular machinery, there are the cytoplasmic molecular chaperones Hsp70 and Hsp90 and their associated co-chaperones, the members of the ubiquitin ligase complex and the ER luminal-interacting protein calnexin folded [9].
Line 286: if two out of three combinations resulted effective in enhancing the QR with respect to
Line 415: Being VX445 used in combination with corrector VX661 and potentiator VX770 in the drug Kaftrio
Line 450: we plan to face the puzzling question that arisen with the use of triple combinations applying our biomolecular approaches to ever smaller CFTR subdomains.
2: The paragraph starting in line 99 is more results than introduction. Please present results in the results section.
The language needs editing. Please see the comments to authors for specific examples.
Reviewer 2 Report
The authors Bongiorno et al performed an excellent study with a high degree of translationality and nicely presented. A few small comments/suggestions to follow before the work is published:
- page 3, lines 105-113: I suggest not anticipating the results obtained in the introduction;
- page 5, line 182: please, indicate abbreviations in full the first time they appear;
- page 7, line 255: do the authors mean Figure 4E? Please, cite 4D figures at line 251.
Reviewer 3 Report
In this manuscript, Bongiorno et al provide an interesting report on the effect of several CFTR correctors upon the expression of isolated CFTR fragments/domains to further characterize the mechanism of action of such compounds, particularly that of VX-445. There are several aspects that need, however, clarification/correction.
Major concerns
1. The most relevant aspect is the detection of full-length CFTR by WB as shown in Fig.1A. Result of such approach is usually a sharp, thin band corresponding to the immature, core-glycosylated form – known as band B - and a diffuse band corresponding to the mature, fully-glycosylated form – known as band C. The pattern shown is exactly the opposite, raising serious concerns that this is not CFTR. As the authors detect both bands even in DMSO-treated cells (and thus cannot use the total absence of band C as a proof that the two-band identification is correct), a proof needs to be shown – the most obvious would be the treatment with Endoglycosidase H and N-glycanase and assess the deglycosylation of immature and mature CFTR.
2. In p.4, lines 150-160, the authors comment on the differential effect of the correctors but statistics seems to have been done only for comparisons with DMSO. To say that the effect is VX-445>VX-809>=VX-661>Corr-4a (as in line 158), statistics needs to be done with all possible pairwise-comparisons.
3. Nomenclature of the domains needs to be coherent throughout the manuscript. Transmembrane domains are named TMD1 and TMD2 in the introduction, but in the title and from the results section onwards they are named MSD1 and MSD2. Unless two different concepts are being evoked, the abbreviations used should be always the same.
4. The authors claim several times that they are testing the effect of correctors upon F508del-CFTR fragments – e.g., in line 102 in the introduction or in line 392 when suggesting that VX-445 binds to MSD2 (but there other similar situations). This is however not true. Unless they are testing NBD1 (where F508 is located), all the other fragments correspond to WT-CFTR fragments. All these sentences need to be rephrased as they are wrong.
5. The lack of effect of VX445 on MSD2 abundance at 8h of CHX chase is puzzling. Can the authors comment on that?
6. Representative tracings should be shown in Figures 2 and 5.
7. In l.410, it is not clear what is meant by “the stability of MSD2 was very low”. How is this comment made, compared to what?
8. The abbreviation HBE is used in the field to refer to primary cultures of human bronchial epithelial cells. In this study, the authors used CFBE-F508del cells which are well known by this name. Please change to make it clearer.
9. No molecular masses are shown in Figure 4.
Minor concerns
10. Please prefer “variants” to “mutations”.
11. Please prefer “people/person with CF (pwCF)” to “CF patients”.
12. In section 2.3, please identify the antibodies used for each fragment (as some are mentioned and others not). In the equivalent “Materials and Methods”, antibodies for each fragment could be shown in a table for increased clarity.
13. The first paragraph of the Discussion is repeating the Introduction and does not contribute to the focus of the current work. Please remove/add any needed detail to the Introduction.
14. In line 426, it is not clear what is meant when the authors say “both VX770 and forskolin”. Weren’t all the assays performed under treatment with both compounds?
15. Please remove the sentences on your future plans or be more specific. It is too vague (lines 448-450).
16. Lines 71 and 72 - use “named” instead of “namely”.
17. Line 73, typo in “on of”.
18. Lines 118 and 133 - use “encoding” instead of “codifying”.
19. Line 145 – typo in “glycosyilated”.
20. Supplementary Table S2 and legend – correct the typos “MDS”.
Round 2
Reviewer 3 Report
The authors addressed most of my questions but the issue of CFTR identity (comment 1) remains.
The description of the differences of the bands is inconsistent - the only reason why band C in rescued F508del-CFTR is thinner than in wt-CFTR is related to the amount that is produced, giving the impression that the glycosylation is not so diverse. As the identity question is not solved, the authors need to show at least wt-CFTR in the same blot (and not just add a lane from a different gel).
Author Response
Comments and Suggestions for Authors
The authors addressed most of my questions but the issue of CFTR identity (comment 1) remains.
The description of the differences of the bands is inconsistent - the only reason why band C in rescued F508del-CFTR is thinner than in wt-CFTR is related to the amount that is produced, giving the impression that the glycosylation is not so diverse. As the identity question is not solved, the authors need to show at least wt-CFTR in the same blot (and not just add a lane from a different gel).
To show the differences among the glycosylation of WT and F508del CFTR, we added a new blot to figure1, showing the expression of WT CFTR obtained from whole cell lysates of HEK-t cells treated with DMSO (control condition) and the four correctors under analysis in this study. As for F508del blots, we loaded also whole cell lysates obtained from cells that were not transfected with any plasmid encoding the CFTR, to be sure that the primary antibody that we used effectively detects its target protein. We also added graphs showing the quantification of total (B + C bands) and mature (bands C/(C+B) ratio) of the WT CFTR protein in each examined condition and commented on the results either in the results and in the discussion sections. As stated in the manuscript, when observing the lanes of control (DMSO treated) whole cell lysate samples, the thickness of the two bands of WT and F508del CFTR results are quite different. As we loaded the same amount of total protein, measured by the Bradford assay, we could say that this difference is due to the different maturation levels that the two isoforms reach into the cells. One of the aims of our study was to understand how much the four correctors that we selected were able to correct the processing defect of F508del, increasing the expression of the F508del mature form to a level closest to that of the WT protein. In Supplementary Table 3, for each examined condition, we added the quantification of the expression of WT CFTR protein, while in Supplementary Table 1, we added the quantification of the mRNA of the WT CFTR detected in our samples by real-time PCR.
Round 3
Reviewer 3 Report
Although WT samples should have be shown in the same gel as F508del ones, the authors answered my concern.